# Parametric evaluation of impedance curve in radiofrequency ablation: A quantitative description of the asymmetry and dynamic variation of impedance in bovine *ex vivo* model

**Ronei Delfino da Fonseca**[1]☯*, **Paulo Roberto Santos**[1]☯, **Melissa Silva Monteiro**[2]☯, **Luciana Alves Fernandes**[3]☯, **Andreia Henrique Campos**[1]‡, **Díbio L. Borges**[4]‡, **Suélia De Siqueira Rodrigues Fleury Rosa**[3]‡

**1** Department of Mechanical Engineering, Technology College, University of Brasília, Brasília, DF, Brazil, **2** Biology Institute, University of Brasília, Brasília, DF, Brazil, **3** Post Graduate Program in Biomedical Engineering—PPGEB, University of Brasília, FGA, Gama, DF, Brazil, **4** Department of Computer Science, University of Brasília, Brasília, DF, Brazil

☯ These authors contributed equally to this work.
‡ These authors also contributed equally to this work.
* ronei@unb.br

## Abstract

Radiofrequency ablation (RFA) is a treatment for liver tumors with advantages over the traditional treatment of surgical resection. This procedure has the shortest recovery time in early stage tumors. The objective of this study is to parameterize the impedance curve of the RFA procedure in an *ex vivo* model by defining seven parameters ($t_{1/2}$, $t_{minimum}$, $t_{end}$, $Z_{initial}$, $Z_{1/2}$, $Z_{minimum}$ and $Z_{end}$). Based on these parameters, three performance indices are defined: one to identify the magnitude of impedance curve asymmetry ($\delta$), one Drop ratio (DR) describing the percentage of impedance decrease until the minimum impedance point is reached, and Ascent Ratio (AR) describing the magnitude of increase in impedance from the minimum impedance point to its maximum point. Fifty ablations were performed in a bovine *ex vivo* model to measure and evaluate the proposed parameters and performance index. The results show that the groups had an average $\delta$ of 29.02%, DR of 22.41%, and AR of 545.33% for RFA without the use of saline or deionized solutions. The saline solution and deionized water-cooled groups indicated the correlation of performance indices $\delta$, DR, and AR with the obtained final ablation volume. Therefore, by controlling these parameters and indices, lower recurrence is achieved.

## Introduction

Hepatocellular carcinoma (HCC) is among the most frequent primary tumors affecting the liver. It is the second most lethal type of cancer [1, 2]. External agents such as hepatitis B and C

**Data Availability Statement:** All relevant data are within the manuscript and its Supporting information files.

**Funding:** Coordenação de Aperfeiçoamento de Pessoal de Nível Superior—Brazil (CAPES).

**Competing interests:** The authors have declared that no competing interests exist.

virus infections, alcoholic cirrhosis, and ingestion of aflatoxins are the main causes of HCC. The 5-year survival rate of HCC patients is 12% [3, 4].

According to Barcelona Clinic Liver Cancer (BCLC), the recommended therapies are transplantation, resection, and local ablative techniques for early stages; however, for the first two options, patients can be ineligible owing to impaired liver function, and only 15%–20% of HCCs are resectable after diagnosis [3, 4]. Therefore, for tumors $\leq 3$ cm radiofrequency ablation (RFA) is still the best therapeutic option available [4]. In other stages, chemical therapies such as percutaneous ethanol injection, transarterial chemoembolization (TACE), or combinations with other methods are used [2]. Ablative therapies are safe, reducing the risks of resection or death in a transplant, thereby increasing patient survival [5].

RFA therapy consists of an alternating radiofrequency current application through a percutaneously inserted electrode in the tumor region [6]. This electrical current agitates the ions adjacent to the electrode and generates heat owing to Joule effect. This heat conducts through the tissue [7]. In addition, irreversible cell damage occurs when the temperature reaches above 60 C [8]. The heat diffusion is limited owing to the dissipative effects of the nearby blood vessel perfusions. The electrical conductivity of the tissue also limits the propagation of heat, since this varies with temperature and water content in tissues. Another factor that directly interferes with the process is the electrical impedance of the tissue. The electrical impedance of biological tissues is influenced by their composition and structure [9]. Behavioral models of the tissue impedance can be obtained using the classic Cole-Cole model [10, 11]. The electrical impedance increases as the tissue is carbonized during the ARF process, which causes an isolation of the electrical circuit formed by the radio frequency generator, active electrode, the target tissue, and the return electrode, leading to a sudden increase in impedance. This fact is cited in the literature as a *roll off* [12–14].

The decrease of impedance during RFA, despite not having a standardized metric, is an indicator of RFA efficiency. Trujillo et al. reported a decrease in impedance when conducting their research [12]. In [15], the researchers tried to evaluate the impact of ablation power and catheter irrigation during the RFA using impedance drop as a parameter. In this study, ablation was guided by varying impedance during the procedure (as a measure of effectiveness) using two models of SmartTouch (ST) and ST surround flow (STSF) catheters. Using a weighted algorithm that considers contact force, power, and duration, experimental evaluation was performed using the full-time force (FTI), cumulative multiple ablation (FTI-P), and ablation index (AI). Authors report that in both catheters, the correlation between the variation of impedance and the AI was present. Ablation through STSF showed lower minimum temperatures at the tip of the catheter; lower impedance drop and stabilized shorter than ST, but the data presented only applies to catheters from the manufacturer used in this study and warns that different irrigation strategies used by other catheter manufacturers may incur different results.

Therefore, the parameter that is viewed as an alternative biophysical measure of the efficacy of the ablative procedure is the impedance variation. However, it is important to highlight the way that the catheter affects the parameters related to the injury [15, 16]. Although the impact of the applied power on the effectiveness of ablation is known, factors affecting it are related to the impedance variation and catheter used.

The studies of Jossinet (1996) with normal and tumorous mammary tissues evaluated the variation of impedance in a frequency range through impedance tomography [17]. The same author, in a subsequent study (Jossinet, 1998) showed that the mammary tissues with carcinoma, obtained through surgical excision, have limited resistivity to low frequency, low fractional power and to the phase angle in frequencies above 125 kHz [18]. A previous study performed by Edd (2005) with rat livers identified impedance as a predictor of the area for freezing when using cryosurgery guided by electrical impedance tomography [19].

Other studies demonstrate that impedance monitoring is an indicator of success in the RFA procedure, reducing the repetition and the procedure duration [20]. Bhaskaran et al. observed the importance of impedance as a parameter for greater efficiency and safety during the cardiac ablation procedure. Using a myocardial phantom, the relationship between a range of impedance (60 Ω, 80 Ω, 100 Ω, 120 Ω, 140 Ω, 160 Ω) and the ablation volume at a fixed power (40 W) was evaluated. In the second experimental set, the power was corrected according to the circuit impedance. During irrigated ablation, the lesion and the overheated dimensions were significantly larger with lower circuit impedance. In contrast, the lesion size was smaller under high impedance conditions. The delivery of energy adjusted to impedance improved the lesion formation consistency and prevented overheating [21].

In the RFA scenario, there is a significant dispersion in the impedance profiles owing to the influence of such factors as the temperature of the tissue under treatment (which is affected by the increased temperature owing to the ablative process), the volume of tissue submitted to the ablative process, and the dissipative effect caused by the presence of blood vessels near the region of treatment.

From the work of Iida et al. who described that there are different impedance curves depending on the type of tumor submitted to RAF [22], this work proposes parameters and performance indexes that allow a quantitative and qualitative analysis of the impedance data and duration of the procedure.

The contributions of this study include the proposition of notable points in the profile of the impedance curve of the RAF as well as a statistically linear model that establishes indices that differ from nonlinear approaches because they are easy to estimate and have no need for the use of robust and/or powerful computational resources.

## Materials and methods

A test was conducted to evaluate the parameters and performance indices involved in the impedance curve for ARF.

To this end, five experimental groups were analyzed: 1—A control group solely entailing the application of radiofrequencies (PURERF); 2—A group with 0.9% saline solution at room temperature (SALINE23); 3—A group with 0.9% saline solution cooled to 5˚C (SALINE5); 4—A group with deionized water at room temperature (DEI23); and 5—A group with deionized water cooled to 5˚C (DEI5).

This assay was conducted in an *ex vivo* setup with bovine liver specimens obtained from a local slaughterhouse; the post mortem was conducted within 30 min at a basal temperature of 37 ˚C. This temperature was maintained constant during the assay. The lobes of each specimen were separated, and standardized samples were obtained with the weight and shape corresponding to a cubic mold with an edge of 6 cm. Ten samples were obtained per group, totaling 50 samples. The anatomical separation of the lobes and selection of samples were based on the Shirai study [23].

All the groups used an ablation equipment developed by the University of Brasilia [24], with a constant supply of 40 W power. The stop criterion was the moment at which the first *roll off* occurred. The Leveen 4.0 (Boston Scientific, Marlborough, MA) semi-open electrode with a diameter of 2.5 cm was used.

The saline solutions and deionized water were applied at a rate of 3 mL/min with application every 4 s. The temperature was monitored using a thermocouple sensor positioned in the center of the Leveen electrode. Fig 1 shows the configuration of the experimental bench used to perform ablations and obtain impedance data. The description of the setup is available on the Protocols.io platform and can be accessed via link [25].

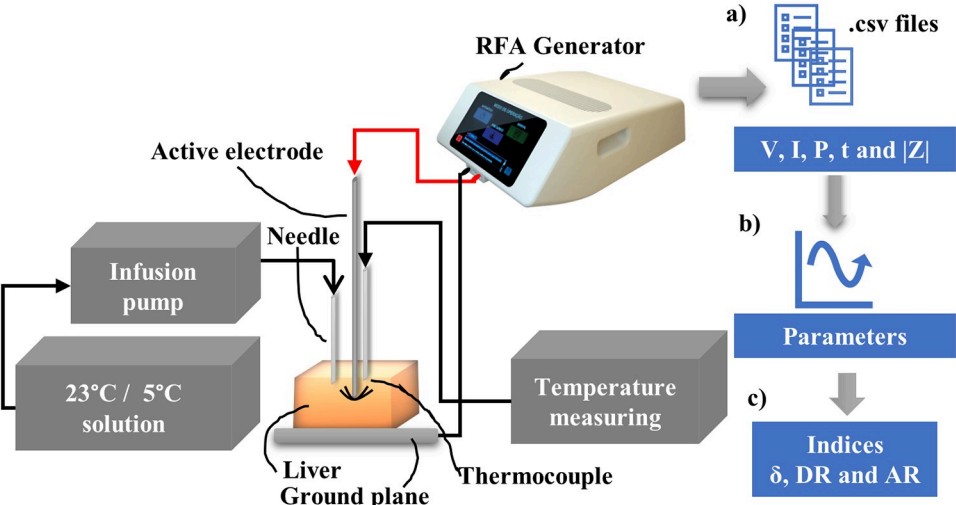

**Fig 1. Experimental setup with the distribution of materials used and measurements for later analyses.** (a) The data on the voltage (V [V]), current (I [mA]), power (P [W]) and time (t [s]) are recorded by the RFA equipment in .*csv* format. (b) The impedance curve parameters are calculated. (c) The indices $\delta$, DR, and AR are estimated.

In this study, to obtain parameters for quantitative and qualitative analyses of the experimental data, we propose seven parameters in the impedance curve, as illustrated in Fig 2, where:

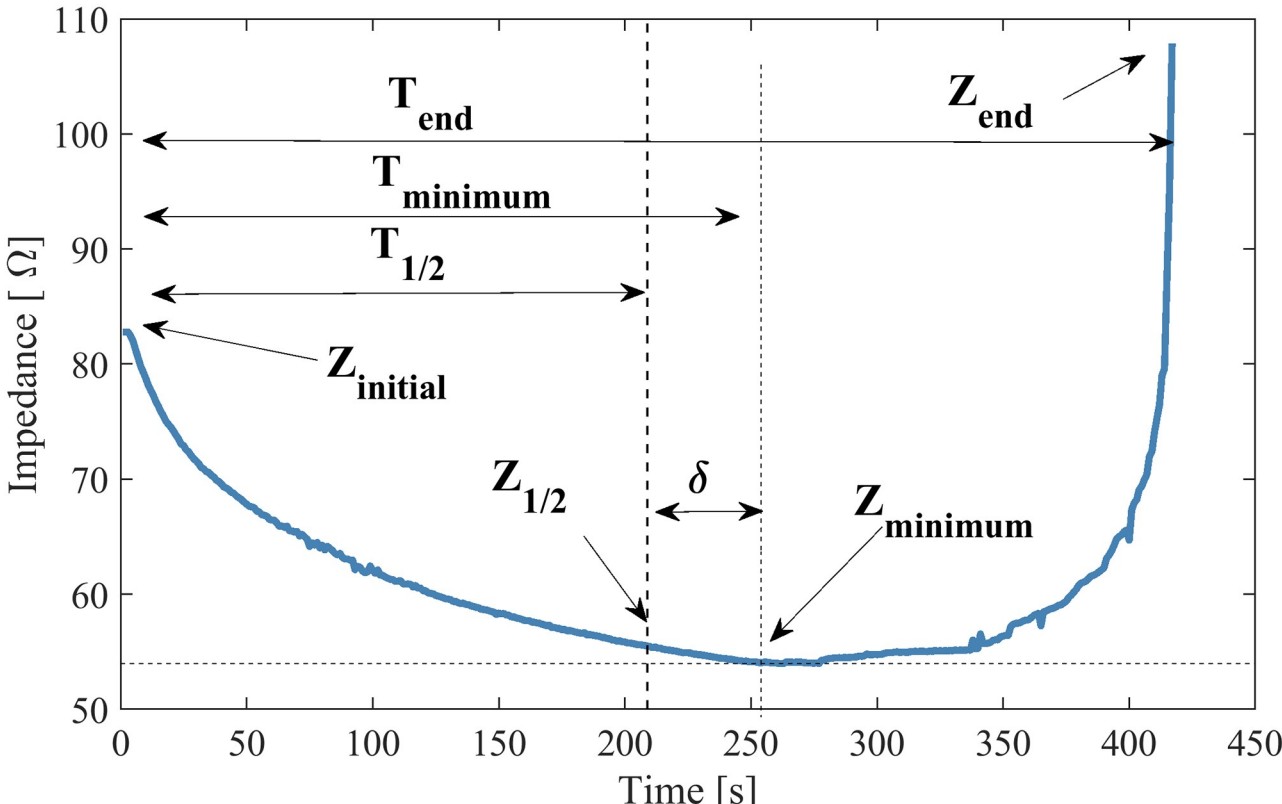

**Fig 2. Impedance parameters.** This figure illustrates time and impedance parameters for an *ex vivo* bovine tissue ablation curve.

1. $t_{1/2}$: Indicates half of the *roll off* duration, $t_{1/2}$ is defined using Eq 1:

$$t_{1/2} = \frac{t_{end}}{2}[s];$$ (1)

2. $t_{minimum}$: The time when the impedance reached the minimum value in the curve [s];

3. $t_{end}$: The duration of the experiment until *roll off* occurs [s];

4. $Z_{initial}$: The initial impedance [Ω];

5. $Z_{1/2}$: The impedance at time $t_{1/2}$ [Ω];

6. $Z_{minimum}$: The minimum impedance in the curve [Ω];

7. $Z_{end}$: The impedance at *roll off* [Ω]. The exact value of this parameter depends on the level where it is considered to have rolled off. We assume the the value of:

$$Z_{end} = 1.5 * Z_{initial}[\Omega];$$ (2)

From these parameters, 3 performance indices are proposed:

1. **Delta ($\delta$):** Measures the distance between $t_{minimum}$ and $t_{1/2}$. It allows evaluating if the minimum impedance point is the same midpoint of the curve. The closer to zero, the closer the points, $t_m$ *inimum* and $t_{1/2}$.

$$\delta = \left(\frac{t_{minimum}}{t_{1/2}} - 1\right) * 100[\%]$$ (3)

2. **Drop Ratio (DR):** Measures the change in impedance between $Z_{initial}$ and $Z_{minimum}$. This index provides an expected estimation of impedance decrease from the initial impedance value to the minimum. DR is given by Eq 4.

$$DR = \left(\frac{Z_{minimum}}{Z_{initial}} - 1\right) * 100[\%]$$ (4)

3. **Ascent Ratio (AR):** Measures the rise of impedance between $Z_{minimum}$ and $Z_{end}$. This index evaluates the difference in impedance at *roll off* from its minimum. AR is given by Eq 5.

$$AR = \left(\frac{Z_{final}}{Z_{minimum}} - 1\right) * 100[\%]$$ (5)

Based on the results obtained from the e*x vivo* experimental sets, comparative analyses were performed between experimental groups based on these performance indices ($\delta$, DR and AR).

## Statistical analysis

As the null hypothesis assumed in this study, the proposed parameters and indices are not influenced when submitted to different types of solutions and temperatures, that is, the average effects of each parameter are equal in each group. This hypothesis leads to the conclusion that the indices, although not exact, represent a range of similar values in each experimental group studied. Thus, it is expected that the p-values in the results will be insignificant. The exception is for the correlation between the indices and volumes of thermal damage.

A statistical analysis was conducted using R software (3.6.0) [26, 27]. The base packages STATS (3.6.0) and LAWSTAT (3.2) of R were used. Shapiro-Wilk test (*shapiro.test*) was used

for the normality test of the variance of the proposed indices, and Levene test (*levene.test*) was used for the homogeneity test.

Two data analysis approaches were conducted: a one-way ANOVA and a two-way ANOVA. The one-way ANOVA was applied to evaluate the behavior of the indices according to the control (PURERF) and experimental (SALINE23, SALINE5, DEIO23, and DEIO5) groups. Based on the statistical response obtained from the ANOVA, a Tuckey HSD post hoc test was used to identify the divergent group.

The two-way ANOVA was applied using two factors (the type of solution used and the temperature submitted) with two levels in each factor (deionized water or a saline solution and an ambient or low temperature) to identify whether these factors and corresponding levels influence the response of the proposed indices and if there is any significant interaction between such factors and levels. Similarly, a Tuckey HSD *post hoc* test was again used to identify the divergent group.

A bootstrapping procedure (1000 re-samples and 95% IC of the corrected and accelerated adjustment) was applied to correct deviations from normality in the samples and ensure the applicability criteria of the ANOVA (normality and homoscedasticity). The procedure consists of randomly taking one of 50 samples from the evaluated indices and including it in a new bootstrap sample group. This sample is then placed in the original group, and the procedure is repeated n times. At the end of n re-samples, the statistics (e.g., confidence intervals and means) are calculated in the created Bootstrap sample [28]. The applications of the method can be seen in several fields of knowledge [29–32]

Nearby outliers were considered as the points that fit the criteria: [*points*] $> \pm 1.5\,^{*}\,IQR$. A correlation study between the volume variable and the performance indices was performed using the Spearman correlation coefficient. A statistical significance of 0.05 was adopted in all tests.

## Results

The macroscopic results of the thermal injury caused and the axes considered in the calculation of volumes are shown in Fig 3. The groups presented different thermal damage. The saline

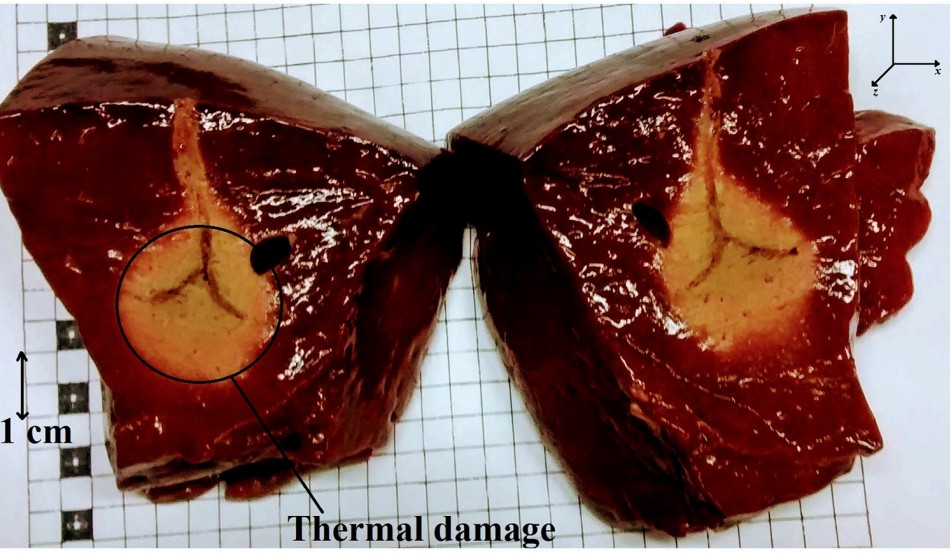

**Fig 3. Ablation result.** The figure illustrates the thermal damage area obtained. The x and y axes were considered for volume calculation, with the z axis being perpendicular to the formed plane.

**Table 1. Summary of volumes obtained by each experimental group.**

| Groups | Volume [cm3] | 95% Confidence Interval | |
|---|---|---|---|
| | mean ± sd | Lower | Upper |
| DEI23 | 13.83 ± 4.73 | 6.06 | 20.0 |
| DEI5 | 14.19 ± 4.20 | 8.59 | 19.7 |
| PURERF | 10.05 ± 2.67 | 6.58 | 13.5 |
| SALINE23 | 16.78 ± 3.79 | 11.4 | 21.6 |
| SALINE5 | 25.08 ± 12.40 | 13.2 | 47.5 |

solution group showed the highest volume of thermal damage. The detailed volume behavior for each group falls short of the scope of this study and has already been published in [33]. Table 1 summarizes the volumes obtained in each experimental group.

## Impedance curves obtained

The RFA impedance curves obtained in each group are illustrated in Fig 4. They present variance in initial, final, and minimum impedance points, and durations. From this obtained data, the parameters and indices were analyzed.

## Performance index $\delta$

The performance index $\delta$ allows for the determination of the midpoint of the RFA procedure duration while assessing whether the minimum impedance point is the same as the midpoint of the curve ($\delta = 0$), thereby determining their asymmetry when these points do not coincide ($\delta \neq 0$). The results showed that $\delta$ is positive for all groups except groups SALINE23 and

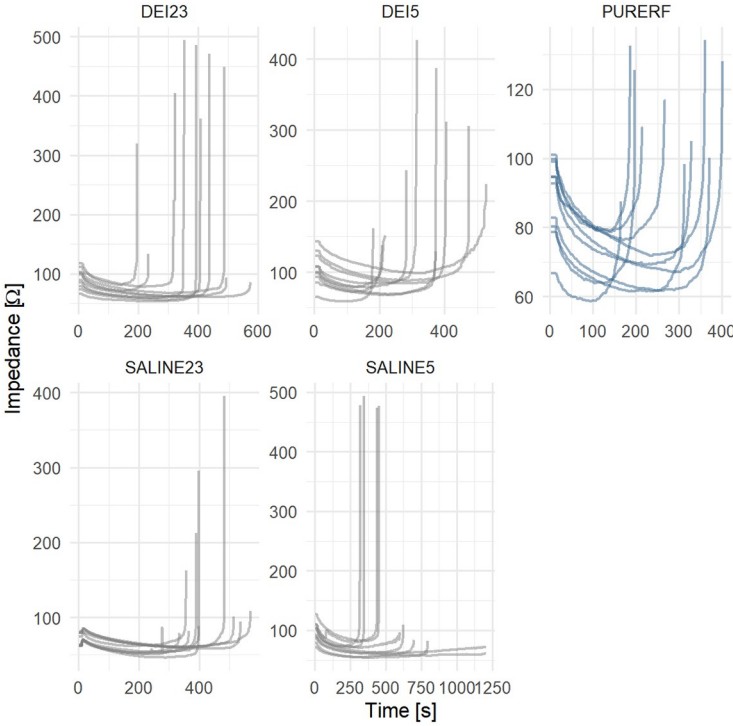

**Fig 4. Impedance curves obtained.** This graph presents the impedance curves of each group for analysis.

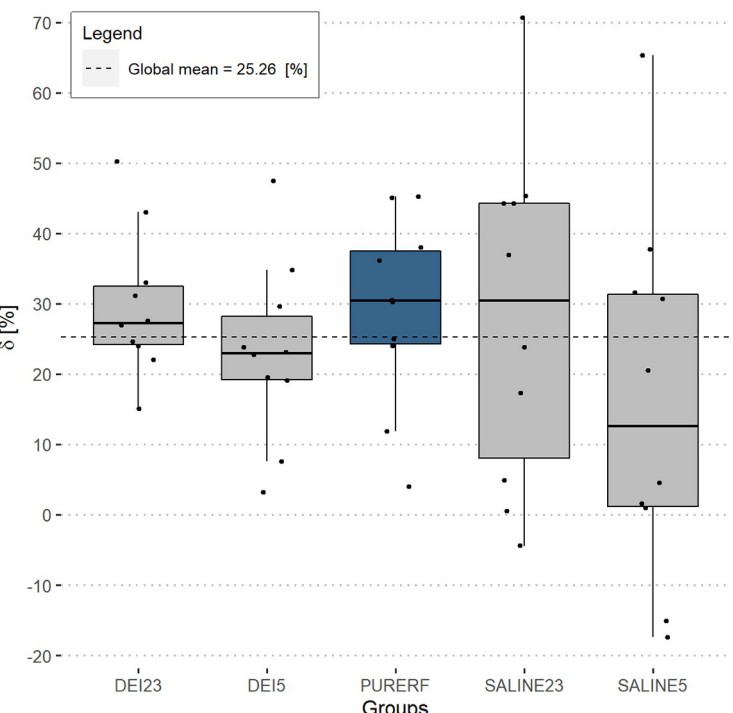

**Fig 5. Performance index δ.** The positive sign of δ indicates two findings: 1—the curve is asymmetric, the procedure midpoint is not the minimum impedance point; and 2—the minimum impedance point is ahead of the RFA operating midpoint. In only two cases did the minimum impedance point occur before the half of experiment time (in the SALINE5 group).

SALINE5, which presented data with 2 negative values (Fig 5). The DEI23 group presented the highest mean among groups (29.8,15.0-50.3,mean,min-max)[%]. The group with minor δ was SALINE5 (16.05,- -17.4–65.4,mean,min-max) [%]. The differences between the evaluated groups are not statistically significant (p = 0.405).

Fig 6 shows a comparison of the correlations between volume and δ. There was no statistical evidence of a correlation between volume and the delta index in any of the groups evaluated. The PURERF control group shows weak positive correlation (R = 0.18, p = 0.63) and the DEI23 group shows negative correlation (R = -0.35 and p = 0.33). However, groups DEI5, SALINE23 and SALINE5 show moderate correlations. The SALINE5 group had the highest negative correlation (R = -0.69 and p = 0.069).

## DR performance index

The DR index indicates the variation between the initial impedance of the tissue compared to the impedance at midpoint of duration. This index presents a mean of 25.8% for all evaluated groups, as shown in Fig 7. The SALINE5 group had the highest DR (31.20,24.2—35.8,mean, min-max) [%] and SALINE23 group had the lowest DR values (21.1,14.3—26.3,mean,min-max) [%]. There was a statistical difference only between SALINE5 group and PURERF group (p = 0.006).

The volume shows signs of negative correlation proportional to the DR index in groups DEI23 and SALINE5. The DEI5 group has a positive and significant correlation (R = 0.8 and p = 0.0059), according to Fig 8.

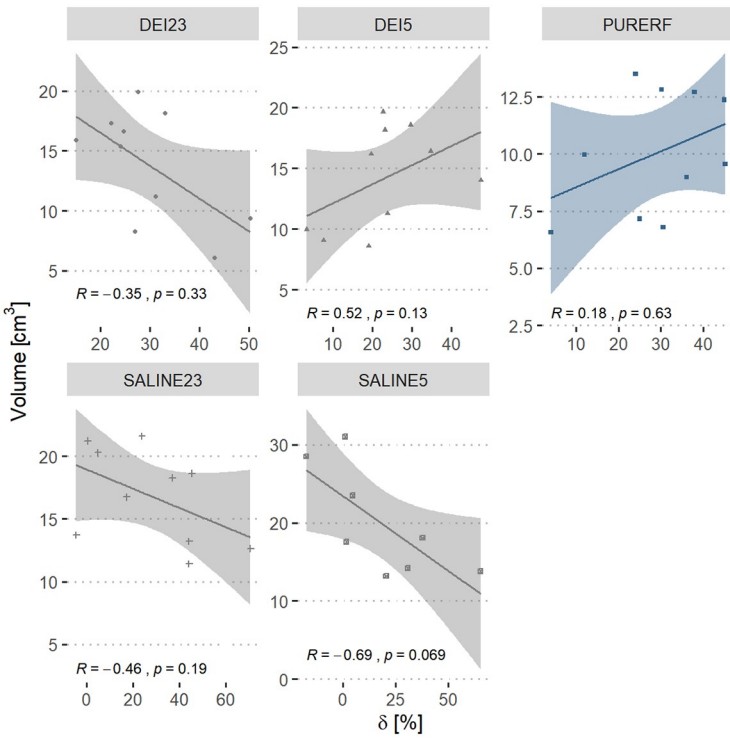

**Fig 6. Correlation between performance index _δ_ and volume.** The graph displays the correlation between volume in the groups and the index _δ_. The PURERF group shows weak evidence of a positive correlation. The SALINE5 group shows the highest evidence of negative correlation (R = -0.69 and p = 0.069).

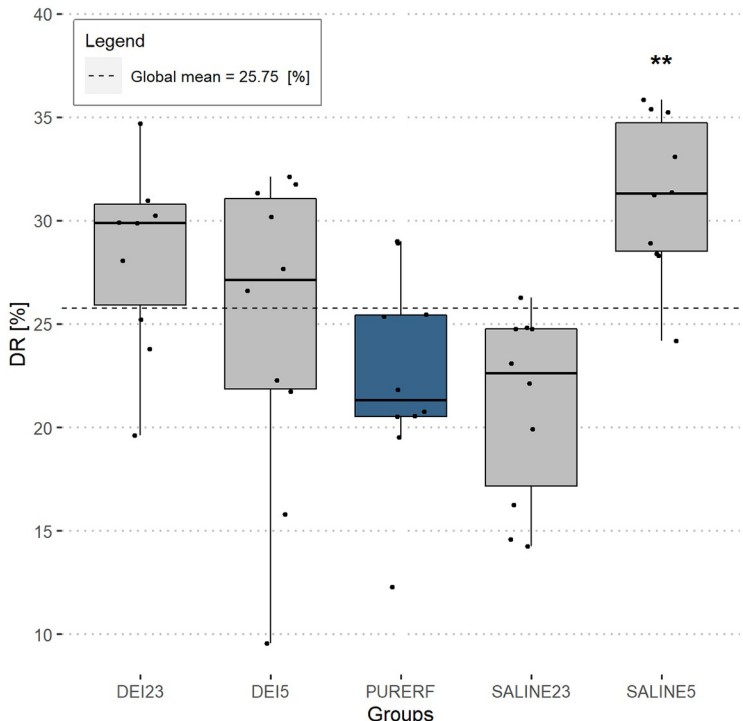

**Fig 7. DR performance index.** The group with saline solution at 5˚C showed the only significant difference compared to PURERF group with a DR of 31.2% and 22.4%, respectively (p = 0.006).

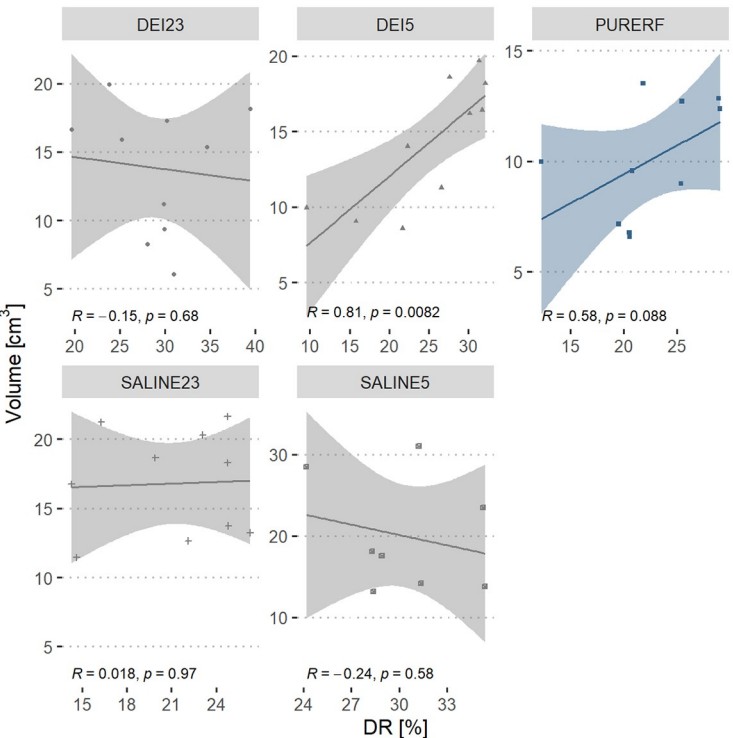

**Fig 8. Correlation between DR performance index and volume.** The graph presents the correlation of the volume in groups to the DR index. The direct correlation is statistically significant in the DEI5 group.

## AR performance index

Measuring the change of impedance from the point where the impedance is minimal to the point where it reaches the *roll off* is done using the AR index. The results show great variability between groups. Differences were detected between groups DEI23 and SALINE23 (p = 0.01), and, PURERF and SALINE23 (p = 0.01), as shown in Fig 9. The SALINE23 group presented lower values of AR compared to the others (167.7,48.4-549.4,mean,min-max,p = 0,002) [%]. The other groups did not present statistical differences among them.

A direct relationship between the AR index and the ablation volume is observed in all groups. This correlation is statistically significant in PURERF (R = 0.72, p = 0.019), DEI23 (R = 0.73, p = 0.015) and SALINE5 (R = 0.78, p = 0.023) groups, as shown in Fig 10.

A two-way ANOVA was conducted to check whether the indices are affected by the solution type (deionized water or saline solution) and the solution temperature (ambient or refrigerated).

Regarding the temperature submitted, the two-way ANOVA did not detect any differences in temperature indications, $\delta$ and AR are not influenced by temperature (p > 0.05). However, when solutions are cooled, more symmetric impedance curves are obtained, demonstrated by the decrease in the mean index of $\delta$ and AR observed in refrigerated groups. The estimated averages are illustrated in Figs 11 and 12, respectively.

The DR indicator was higher at the refrigerated level compared to the PURERF level (p = 0.028), with averages of 28.04 and 22.41, as illustrated in Fig 13.

There was significant interaction between the factors (groups and temperature) for the indices, DR and AR (p = 0.001). The interaction observed in Figs 12 and 13 indicate a point where DR and AR are equal for saline solutions and deionized water.

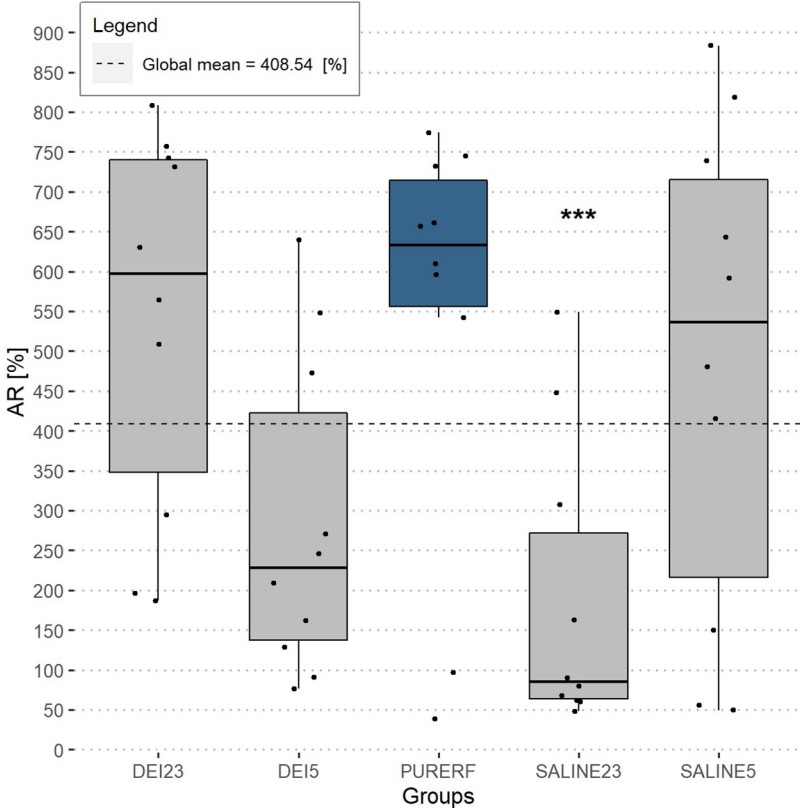

**Fig 9. AR performance index.** The AR index detects the variation between the minimum point of impedance to the point of occurrence of the first *roll off*. All experimental groups resulted in accentuated AR, except for SALINE23 with (167.7,48.4-549.4,mean,min-max,p = 0.002). Only SALINE23 showed a statistical difference, compared to the PURERF.

In the case of solution type, there were no differences.

Tables 2 and 3 present the estimated means and their respective confidence intervals for each proposed index.

## Discussion

The performance index $\delta$ shows a direct relationship to the volume. Greater the asymmetry in the impedance curve (indicated by larger $\delta$), larger the final volume likely obtained. This is partially owing to the fact that larger $\delta$ implies a longer ablation time with the displacement of *roll off* leading to a delay in the carbonization of the electrode and a greater energy deposition in the tissue. Therefore, $\delta$ indicate the extend of thermal damage. No statistical differences were evident when comparing RFA, saline and deionized solutions, which indicate that $\delta$ is independent of the employed solution type. There was a decrease in $\delta$ by cooling, even though no statistical differences were detected.

During the RFA procedure, the impedance curve of the tissue changes in magnitude as the tissue undergoes carbonization and water loss. In this study, it was observed that the initial impedance was at a level of (66–117 $\Omega$) and decreased around 29% on average for the PURERF group, as verified by DR (Tables 1 and 2). This impedance drop is associated to the change in the electrical resistivity of the tissue.

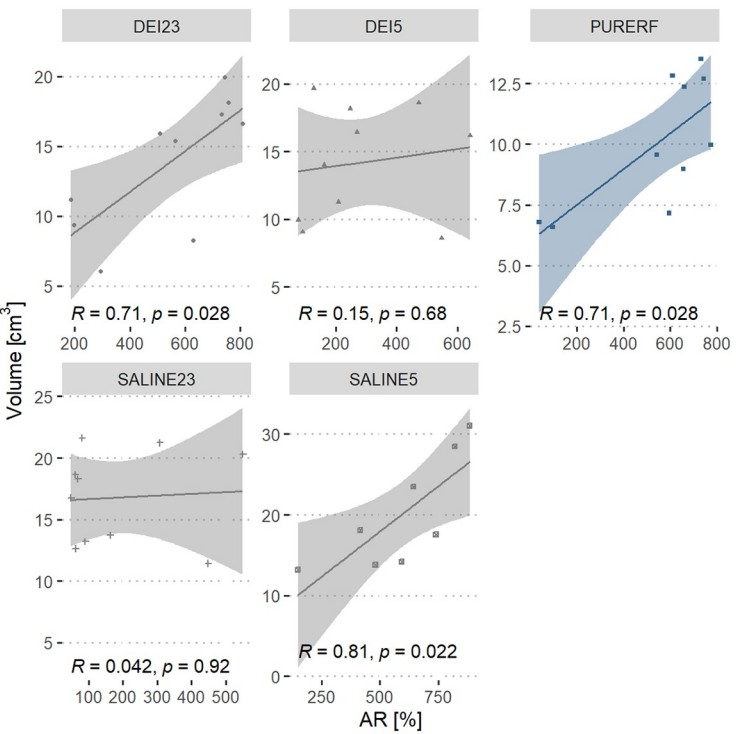

**Fig 10. Correlation between AR performance index and volume.** The graph presents the correlation between the volume in the groups and the AR index.

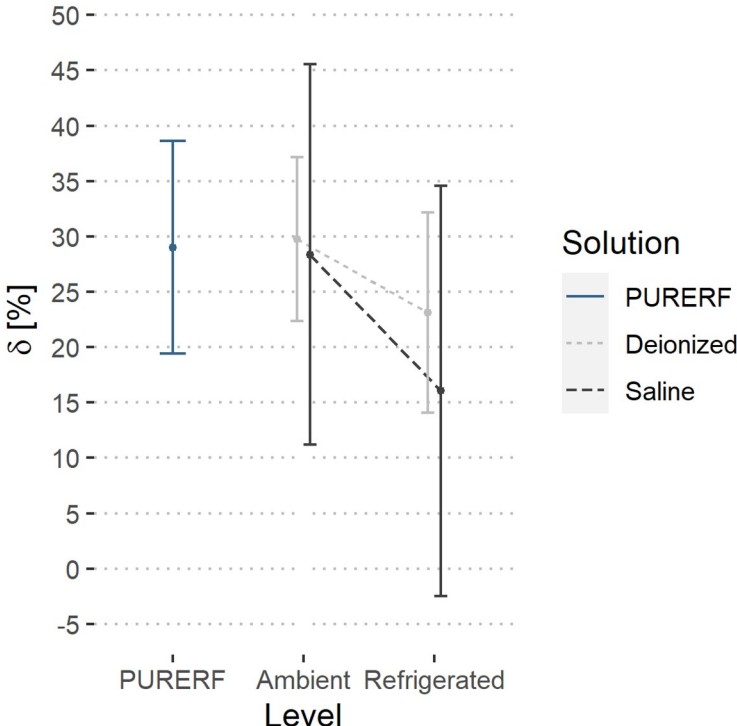

**Fig 11. Estimated marginal averages of the index δ.** The graph indicates no significant differences owing to the solution type or the temperature. Furthermore, there were no interactions between the factors.

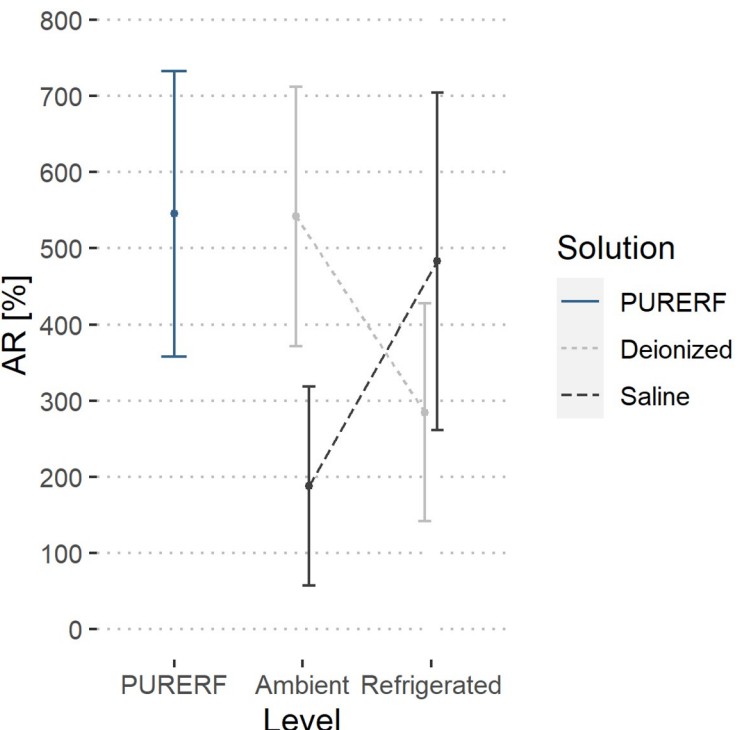

**Fig 12. Estimated marginal averages of the AR indicator.** The graph indicates no significant interactions for the solution type or the temperature. The AR index decreases in both solutions when refrigerated. This indicates a lower final impedance.

The electrical resistivity of tissues has a non-linear behavior and is temperature dependent. The behavior of the resistivity can be modeled in parts using linear and exponential functions as demonstrated by [34]. In general, the dynamics of the resistivity decreases with increasing temperature up to a threshold between 75°C and 100°C, when non-linear behavior occurs [34]. That is, as heat is generated by the electrode, the resistivity is affected until the minimum impedance value ($Z_{minimum}$) is reached. Here, the impedance increases. This reversal in impedance behavior occurs owing to the change in the inclination of electrical resistivity when it reaches the non-linear growth region [34] and is accentuated by the tissue degradation while heated. This tissue degradation isolates the electrode and prevents heat propagation to more distant zones, thereby limiting the area of lethal damage [6].

The DR signals the drop in impedance owing to electrical resistivity changes of the tissue during the entire temperature excursion in the RFA process. Therefore, DR includes the part of the resistivity that decreases with an increase in temperature, which is often approximated using linear functions. When performing RFA, it is expected that the impedance will decrease over time. However, the degree of decrease in impedance is rarely reported in the literature. Trujillo et al., while studying the *roll off* phenomenon noticed that, on average, the impedances fell around 27.0% in their experiments on *ex vivo* bovine model, similar to the one obtained in this study with deionized solution that presented a mean DR of 27.04% (Table 2). The saline solution when cooled allows obtaining a higher impedance drop compared to RFA. The DR performance index can be an indicator of the degree of decrease in initial impedance during a successful RFA procedure.

Likewise, it is desirable to know the impedance behavior once it reaches the minimum point and begins to rise gradually until the *roll off* event. This can be measured using AR,

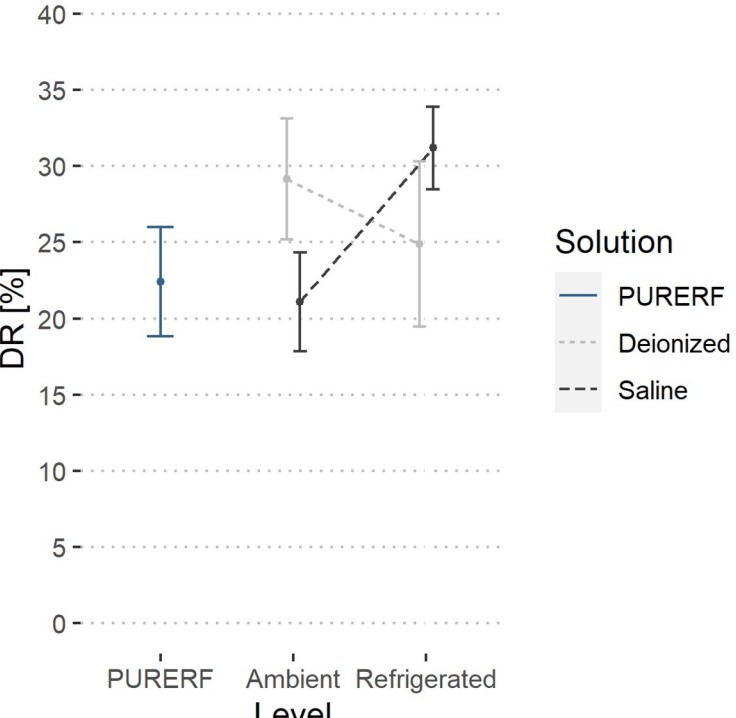

**Fig 13. Estimated marginal averages of the DR indicator.** The graph shows the interaction between the solution type and temperature. An inverse behavior between the solution type is observed with the change in temperature. The saline solution allows for a higher impedance drop compared to RFA.

**Table 2. Estimated marginal averages of performance indices for groups DEI23, DEI5, PURERF, SALINE23 and SALINE5.**

| Index | Level | Mean | 95% Confidence Interval | |
|---|---|---|---|---|
| | | | Lower | Upper |
| $\delta$ | DEI23 | 29.76 | 18.04 | 41.48 |
| | DEI5 | 23.1 | 11.38 | 34.82 |
| | PURERF | 29.02 | 17.3 | 40.73 |
| | SALINE23 | 28.36 | 16.64 | 40.08 |
| | SALINE5 | 16.05 | 4.33 | 27.77 |
| DR | DEI23 | 29.18 | 25.71 | 32.64 |
| | DEI5 | 24.9 | 21.43 | 28.37 |
| | PURERF | 22.41 | 18.94 | 25.88 |
| | SALINE23 | 21.09 | 17.62 | 24.56 |
| | SALINE5 | 31.19 | 27.72 | 34.66 |
| AR | DEI23 | 542.13 | 387.56 | 696.69 |
| | DEI5 | 284.6 | 130.03 | 439.16 |
| | PURERF | 545.33 | 390.76 | 699.89 |
| | SALINE23 | 187.67 | 33.11 | 342.24 |
| | SALINE5 | 482.96 | 328.4 | 637.53 |

**Table 3. Estimated marginal averages of the performance indices for the factors solution type and temperature level.**

| | Factor | Level | Estimate | 95% Confidence Interval | |
|---|---|---|---|---|---|
| | | | | Lower | Upper |
| $\delta$ | Solution | DEIONIZED | 26.43 | 18.07 | 34.78 |
| | | PURERF | 29.02 | 17.2 | 40.83 |
| | | SALINE | 22.21 | 13.85 | 30.56 |
| | Level | AMBIENT | 29.06 | 20.89 | 37.23 |
| | | REFRIGERATED | 19.58 | 11.41 | 27.74 |
| | | PURERF | 29.02 | 17.47 | 40.57 |
| DR | Solution | DEIONIZED | 27.04 | 24.15 | 29.92 |
| | | PURERF | 22.41 | 18.33 | 26.49 |
| | | SALINE | 26.14 | 23.25 | 29.03 |
| | Level | AMBIENT | 25.13 | 22.3 | 27.96 |
| | | REFRIGERATED | 28.05 | 25.22 | 30.88 |
| | | PURERF | 22.41 | 18.41 | 26.41 |
| AR | Solution | DEIONIZED | 413.36 | 292.06 | 534.67 |
| | | PURERF | 545.33 | 373.78 | 716.88 |
| | | SALINE | 335.32 | 214.01 | 456.62 |
| | Level | AMBIENT | 364.9 | 242.58 | 487.22 |
| | | REFRIGERATED | 383.78 | 261.46 | 506.1 |
| | | PURERF | 545.33 | 372.34 | 718.31 |

which suggests the expected rise in impedance, in view of the value obtained at the minimum point of the impedance curve. In this study, AR presents an average of 545% when no solutions are used, i.e., the final impedance obtained in the first *roll off* is about 5.5 times the minimum impedance. The magnitude of AR decreased when saline or deionized solutions were used at the studied temperature levels.

Altogether, the performance indicators $\delta$, DR and AR allow for an objective analysis of the impedance behavior during the RFA procedure. The symmetry of the curve and consequently, the middle point of the RFA procedure are observed using $\delta$. The DR allows us to identify if the minimum impedance is achieved and finally, AR tells us the degree of increase in impedance when reaching the first *roll off*.

## Conclusion

Parameterization of the impedance curve in RFA provides data for an objective process evaluation. The performance index $\delta$ indicates the symmetry of the impedance curve and the middle point of the process. For evaluated groups, $\delta$ presented an average of 29.02% for PURERF. The DR index provides an estimate of the expected decrease in impedance and presents an average of 22.41%. The AR index shows that in this setup, the final impedance is 5.5 times higher than the minimum impedance.

In general, these indices showed few significant differences between the evaluated groups. Therefore, the performance indices vary little with the solution type (saline or deionized water) or the solution temperature (ambient or cooled). A particular case is DR that presents the interactions between the solution type and its temperature. Therefore, these indices can be used as performance indicators of RFA to determine its efficiency.

The limitations of this study is the non-validation using *in vivo* experiments, which will certainly bring different values in scenarios that include dynamics, which are not considered in

this study, such as the dissipative effects owing to blood perfusion of nearby vessels, which are beyond the clinical status of the patient.

## Supporting information

**S1 File. Dataset.**
(XLSX)

## Acknowledgments

The present work was carried out with the support received from the Postgraduate Programs in Mechatronics Engineering (PPMEC), Biomedical (PPGEB), the Institute of Biological Sciences (IB) of the University of Brasilia and the Federal Institute of Rondônia (IFRO).

## Author Contributions

**Conceptualization:** Ronei Delfino da Fonseca.

**Formal analysis:** Ronei Delfino da Fonseca, Andreia Henrique Campos.

**Investigation:** Paulo Roberto Santos, Melissa Silva Monteiro, Luciana Alves Fernandes.

**Methodology:** Paulo Roberto Santos, Melissa Silva Monteiro, Luciana Alves Fernandes, Suélia De Siqueira Rodrigues Fleury Rosa.

**Project administration:** Andreia Henrique Campos, Suélia De Siqueira Rodrigues Fleury Rosa.

**Software:** Luciana Alves Fernandes, Díbio L. Borges.

**Supervision:** Ronei Delfino da Fonseca, Díbio L. Borges.

**Validation:** Luciana Alves Fernandes, Andreia Henrique Campos, Díbio L. Borges, Suélia De Siqueira Rodrigues Fleury Rosa.

**Visualization:** Díbio L. Borges.

**Writing – review & editing:** Paulo Roberto Santos, Melissa Silva Monteiro, Andreia Henrique Campos.

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
