## [Editor Report · Decision Letter 0]

29 Jul 2020

PONE-D-20-20867

Parametric evaluation of the resistance curve in Radiofrequency Ablation: a quantitative description of the asymmetry and dynamic variation of resistance in an ex vivo bovine model

PLOS ONE

Dear Dr. Ronei Delfino da Fonseca,

Thank you for submitting your manuscript to PLOS ONE. After careful consideration, we feel that it has merit but does not fully meet PLOS ONE’s publication criteria as it currently stands. Therefore, we invite you to submit a revised version of the manuscript that addresses the points raised during the review process.

I read your paper by myself and found that it's not ready for sending to reviewers. I could initiate the review after you fully address my concerns found below.

We look forward to receiving your revised manuscript.

Kind regards,

Eugene Demidenko, Ph.D.

Academic Editor

PLOS ONE

Additional Editor Comments:

While characterization of electrical properties of bovine liver is an important task there are several critical issues and concerns the authors should address before initiating the review process. Below are my major concerns:

1. The authors should use the term electrical resistance and impedance because there are several meanings of resistance.

2. The authors have a mix of the terms “resistance” and “impedance.” It is common to use resistance for DC and impedance for AC current. The word “resistance” is in the title but the y-axis in Fig. 2 is labeled “Impedance.”

3. The electrical impedance of ex vivo human tissues were comprehensively studied in previous literature: See: Jossinet J (1996). Variability of impedivity in normal and pathological breast tissue. Med.& Biol. Eng. & Comput. 34:346—350 and Jossinet J (1998). The impedivity of freshly excised human breast tissue. Physiological Measurements 19:61—75. The authors miss these important references.

4. Fig. 2 shows only one curve how impedance depends on time. I suggest showing all curves in five plots corresponding to 5 groups. How easy is to detect the points on the curve displayed in Fig. 2? My impression is that in some curves the identification of those points is not straightforward and therefore this method is prone to large variation and imprecise (see the next item).

5. In general, I’m not impressed with the suggested method of characterizing the impedance of bovine liver due to high scatter: most results are not statistically significant. I suggest the authors to investigate two approaches to improve the comparison across groups: (a) two-way ANOVA when all points on the curve are used simultaneously for group discrimination, (b) nonlinear modeling approach as outlined in ref [20].

Journal Requirements:

3. Please ensure that in your methods section you have provided details of the sources of all materials, chemicals, equipment and instrumentation used in your study, including the bovine livers. This is in line with our reproducibility criterion for publishing, see https://journals.plos.org/plosone/s/criteria-for-publication#loc-3.

"R.D.F, A.H.C, P.R.S, S.S.R.F.R

Dean of Research and Innovation (DPI)

http://dpi.unb.br/en/

NO"

"This research was funded by the Dean of Research and Innovation (DPI) of

the University of Brasilia (UnB) and by the Graduate Program in

Mechatronic Systems (PPMEC) of the University of Brasilia."

---

## [Author Response · Author response to Decision Letter 0]

21 Oct 2020

Response to Additional Editor Comments:

(Editor) While characterization of electrical properties of bovine liver is an important task there are several critical issues and concerns the authors should address before initiating the review process. Below are my major concerns:

1. The authors should use the term electrical resistance and impedance because there are several meanings of resistance.

(Answer ): We are grateful for the suggestion of the possible application of both terms, considering that only when using the real part in the analysis the term resistance can be used, that is, it is treated as purely resistive. However, according to the more detailed explanation below, the group opted to standardize the use of the term impedance in the manuscript.

2. The authors have a mix of the terms “resistance” and “impedance.” It is common to use resistance for DC and impedance for AC current. The word “resistance” is in the title but the y-axis in Fig. 2 is labeled “Impedance.”

A: We welcomed the suggestion to standardize the term impedance in the study applied to RFA. Impedance is composed of a real and imaginary part, the latter being the reactive component in the system. From the radiofrequency point of view, there are two elements: a purely resistive element associated with a capacitive reactance. This indicates that there is a phase shift between the voltage and current applied during the RFA. When using only the real part in the analysis, the term resistance can be used, that is, it is treated as purely resistive. However, in this case it is consistent to use impedance in agreement with the application of the RFA technique with alternating current, even though only the purely resistive part, sufficient to parameterize satisfactory performance indexes for clinical application, is analyzed. Thus, we have readjusted the title of the manuscript so that it is consistent with what has already been established in literature.

3. The electrical impedance of ex vivo human tissues were comprehensively studied in previous literature: See: Jossinet J (1996). Variability of impedivity in normal and pathological breast tissue. Med.& Biol. Eng. & Comput. 34:346—350 and Jossinet J (1998). The impedivity of freshly excised human breast tissue. Physiological Measurements 19:61—75. The authors miss these important references.

A: We are grateful for the suggestion to include the reference of the study carried out on normal and tumor human tissues, in addition to others, allowing an overview of the basic parameters for impedance assessment. With these data, there was an enrichment of the text about the basic concepts on impedance in human tissues present in literature, in addition to those existing in liver tissues already brought in the text and more current. The aforementioned author carried out a comprehensive assessment regarding the behavior of the variable in a frequency range and with diversity of tissues for comparison.

4. Fig. 2 shows only one curve how impedance depends on time. I suggest showing all curves in five plots corresponding to 5 groups. How easy is to detect the points on the curve displayed in Fig. 2? My impression is that in some curves the identification of those points is not straightforward and therefore this method is prone to large variation and imprecise (see the next item).

A: We appreciate the suggestion to include the graph of the impedance profiles. They actually made it clear what the profile is in each group. As suggested, the group added a subsection “Obtained impedance curves” in the results section and another graph (Fig. 4) showing the impedance profile divided by group. In fact, there is a great variability of the curves that is expected in RFA, this dispersion being characteristic of the RFA process. Several factors contribute to this, such as the temperature of the tissue being treated, which is affected by the temperature rise caused by the ablation, the volume of tissue submitted to the ablative process, and the dissipative effect caused by the presence of blood vessels, among other factors. Thus, it is a challenge to obtain a model that incorporates all these dynamic variables. The proposed analysis aims to allow the clinician a preliminary view based on simple parameters to be calculated.

Contrary to the first impression obtained that the points are not easy to identify, the curves shown allow the visualization and determination of the points for calculating the indices in each evaluated group.

5. In general, I’m not impressed with the suggested method of characterizing the impedance of bovine liver due to high scatter: most results are not statistically significant. I suggest the authors to investigate two approaches to improve the comparison across groups: (a) two-way ANOVA when all points on the curve are used simultaneously for group discrimination, (b) nonlinear modeling approach as outlined in ref [20].

 A: Thank you very much for the comment. The dispersion of the data is something that cannot be eliminated as explained in the previous point, being a characteristic of RFA. The group took great care in minimizing dispersions and biases in the data, either in the experimental execution or in the treatment of the data during the analyses, evaluating the influences of outliers in the general result.

 The null hypothesis of this work is that the proposed parameters and indices are not influenced when subjected to different types of solutions and temperatures, that is, the means of the effects are the same in each group. Thus, it is expected that in the results the p-values are not significant. The exception is for the correlation between indices and volumes.

a) As suggested, we performed a two-way ANOVA with the factors being the type of solution (SALINE, DEIONIZED and the RFA control) and the temperature of the solution (ROOM TEMPERATURE, REFRIGERATED and the RFA control). The results corroborate with the analyses carried out in the previous manuscript and demonstrate that the indices, despite the dispersion of the data, present similar values in the analyzed factors. These findings were incorporated into the manuscript along with 3 new images (Figs 11, 12 and 13) and Table 1 was revised, which now points to the mean estimates and their confidence intervals instead of the previous minimum and maximum values, in addition to a new one table for the analyzed factors (Table 2). 

b) The suggestion to use a non-linear approach is a great one and has already been carried out by the group in a previously published work. Unlike the nonlinear approach that seeks to describe the impedance dynamics as a function of the procedure time, this work seeks to standardize notable points on the impedance curve. We believe that the present work is different because it is easy to estimate without using computational resources.

---

## [Decision Letter · Decision Letter 1]

12 Nov 2020

PONE-D-20-20867R1

Parametric evaluation of impedance curve in radiofrequency ablation: A quantitative description of the asymmetry and dynamic variation of impedance in bovine ex vivo model

PLOS ONE

Dear Dr. Ronei Delfino da Fonseca,

Thank you for submitting your manuscript to PLOS ONE. After careful consideration, we feel that it has merit but does not fully meet PLOS ONE’s publication criteria as it currently stands. Therefore, we invite you to submit a revised version of the manuscript that addresses the points raised during the review process.

The paper has been improved since the original submission and the authors addressed my critique. However, the reviewer made several constructive comments and I suggest you to address those carefully. In particular, I agree with insufficient information provided in Statistical Methods section. For example, the two-way ANOVA model and bootstrapping were not described at the level of complete understanding. What base R functions or statistical packages have been used and what the outcome of those testing/estimation have been?

I could accept the paper only if you fully address those comment.

We look forward to receiving your revised manuscript.

Kind regards,

Eugene Demidenko, Ph.D.

Academic Editor

PLOS ONE

Reviewers' comments:

Reviewer's Responses to Questions

**Comments to the Author**

1. If the authors have adequately addressed your comments raised in a previous round of review and you feel that this manuscript is now acceptable for publication, you may indicate that here to bypass the “Comments to the Author” section, enter your conflict of interest statement in the “Confidential to Editor” section, and submit your "Accept" recommendation.

Reviewer #1: (No Response)

2. Is the manuscript technically sound, and do the data support the conclusions?

Reviewer #1: Yes

3. Has the statistical analysis been performed appropriately and rigorously? 

Reviewer #1: No

4. Have the authors made all data underlying the findings in their manuscript fully available?

Reviewer #1: Yes

5. Is the manuscript presented in an intelligible fashion and written in standard English?

Reviewer #1: Yes

6. Review Comments to the Author

Reviewer #1: General Comment:

This paper presents an attractive impedance-based method for evaluating the radiofrequency ablation in biological tissue. The authors address the problem by parameterizing the impedance measurements using three performance indices relating the sample conditions. The paper details the experimental conditions and presents useful results based on experimental data. The paper has an adequate structure is mostly clear. I feel that the paper has value to be published by fixing some quantitative and qualitative details. Some suggestions are proposed to improve the paper readability and quality.

Comment 1:

Line 43. The authors claim the impedance of the tissues and a circuit. There are some observations here.

1) Clarify that you are talking about electrical impedance.

2) Formally, the impedance is not an intrinsic electrical property of the tissue as the conductivity and permittivity. In such a sense, the impedance is a generalized property that derives from effective conductivity and permittivity from multiple layers (fat, blood, water. etc). It will be useful to clarify this situation by adding a reference (eg. https://doi.org/10.1016/j.electacta.2018.04.167).

3) What circuit does the authors are referring? Is it an equivalent circuit model? Again, this latter is a macroscopic approximation of the effective conductivity and permittivity. You could use the same reference of the comment above.

Comment 2:

Line 62. Typo: “though” instead of “through”.

Comment 3:

Line 86. Use italic font for “ex vivo”.

Comment 4:

In Figure 1. The label for the temperature sensor is incorrect, should be “thermocouple” instead of “thermopar”.

Comment 5:

In Methods section. The authors talk about the experimental setup and equipment. It is not clear, at least for me, how the impedance measurement was done? By inspection of Figure 1, there is solely one electrode coming from the RFA. The authors should explain briefly, in the manuscript, which is the transducing mechanism for the impedance readings, owing that is a key concept of the paper.

Comment 6:

In the abstract, the word “fall” is missing for the parameter (ROF).

Comment 7:

In Equations (2), (3) and (4) the subscript “minimum” has a typo, it says "mínimum" (accent). Please, thoroughly revise the whole manuscript for typos.

Comment 8:

The caption of Figure 3 talks about x,y,z axes, however in the Figure itself they don´t appear. Add a reference coordinate system in the Figure to improve the readability.

Comment 9:

Due to the work presents a lot of statistical analysis from experimental data. It will be quite useful for the readership to extent the Statistical Analysis section. Instead of only set forth which software and methods are used, the authors should explain in detail why and how the statistical methods were selected.

Comment 10:

In the caption of Figure 5, the variable is missing after the phrase “Performance index”. Moreover, the last sentence is difficult to read.

Comment 11.

The sentence in line 194 has grammatical errors. Check spelling, grammar and typos along the whole manuscript.

Comment 12.

In the Discussion, the ideas therein are certainly true; however, it shows some mixed concepts and unclear ideas. For instance,

1) Up to this point of the paper, the authors had not introduced neither linear nor non-lineal modeling of the electrical resistivity. It thus could confuse the readership owing that the work is supposed to be based on “linear” modeling by statistics.

It will be quite useful, again, to mention why they used linear statistical models instead of non-linear.

7. PLOS authors have the option to publish the peer review history of their article (what does this mean?). If published, this will include your full peer review and any attached files.

Reviewer #1: No

---

## [Author Response · Author response to Decision Letter 1]

17 Dec 2020

Dear Reviewer(s),

It is a great satisfaction for our group to forward this work after careful analysis and corrections.

We appreciate the relevant comments made to improve the work done. We inform you that all suggestions for changes and additions have been accepted and incorporated into the text.

We are available to address any other issues that may arise.

Regards,

Reviewer #1: General Comment:

This paper presents an attractive impedance-based method for evaluating the radiofrequency ablation in biological tissue. The authors address the problem by parameterizing the impedance measurements using three performance indices relating the sample conditions. The paper details the experimental conditions and presents useful results based on experimental data. The paper has an adequate structure is mostly clear. I feel that the paper has value to be published by fixing some quantitative and qualitative details. Some suggestions are proposed to improve the paper readability and quality.

Comment 1:

Line 43. The authors claim the impedance of the tissues and a circuit. There are some observations here.

1) Clarify that you are talking about electrical impedance.

A: Thank you for the suggestion. We have completely modified the passage mentioned. We have added a description of the tissue impedance and provided references.

2) Formally, the impedance is not an intrinsic electrical property of the tissue as the conductivity and permittivity. In such a sense, the impedance is a generalized property that derives from effective conductivity and permittivity from multiple layers (fat, blood, water. etc). It will be useful to clarify this situation by adding a reference (eg. https://doi.org/10.1016/j.electacta.2018.04.167).

A: We greatly appreciate your relevant comment. We have fully accepted your suggestion and rewrote the entire excerpt including the valuable reference provided.

3) What circuit does the authors are referring? Is it an equivalent circuit model? Again, this latter is a macroscopic approximation of the effective conductivity and permittivity. You could use the same reference of the comment above.

A: We appreciate your comment and suggestion. We clarify that the mentioned circuit corresponds to the electrical system formed by the RF generator, active electrode, the target tissue, dermis, epidermis, muscles, and the return electrode, as shown in Figure 1 [5]. Thus, the impedance that is obtained by the RFA device is the equivalent impedance of the elements of this electrical circuit. As this is an ex vivo study, the influences of the dermis, epidermis and muscles were disregarded.

Fig 1. Fonte: RFA procedure [5].

Comment 2:

Line 62. Typo: “though” instead of “through”.

A: We appreciate the comment. We sent the manuscript for professional spell checking and the term was corrected. We also carefully reviewed the manuscript for other typographical errors.

Comment 3:

Line 86. Use italic font for “ex vivo”.

A: We appreciate the comment. The term is now presented in italics as suggested.

Comment 4:

In Figure 1. The label for the temperature sensor is incorrect, should be “thermocouple” instead of “thermopar”.

A: We appreciate the comment. That was a translation error. The correct term was added to the image.

Comment 5:

In Methods section. The authors talk about the experimental setup and equipment. It is not clear, at least for me, how the impedance measurement was done? By inspection of Figure 1, there is solely one electrode coming from the RFA. The authors should explain briefly, in the manuscript, which is the transducing mechanism for the impedance readings, owing that is a key concept of the paper.

A: Your comment was very helpful and we are grateful that you have noticed this error in the description of the impedance measurement. In fact, this is a key point that was not described in the manuscript. Now, we give details of this measurement made internally by the equipment using the active electrode and the return electrode. The equipment provides the parameters: applied voltage, current, impedance and duration of the procedure in a .csv file (comma-separated values). The parameters and respective performance indexes were calculated based on this file. We also remade Figure 1 of the experimental setup to better illustrate the procedure.

Comment 6:

In the abstract, the word “fall” is missing for the parameter (ROF).

A: We appreciate the comment and inform you that entire manuscript has been sent back to a specialized translation service.

Comment 7:

In Equations (2), (3) and (4) the subscript “minimum” has a typo, it says "mínimum" (accent). Please, thoroughly revise the whole manuscript for typos.

A: We appreciate the comment and inform you that the authors have reviewed the manuscript thoroughly in search of errors of this nature.

Comment 8:

The caption of Figure 3 talks about x,y,z axes, however in the Figure itself they don´t appear. Add a reference coordinate system in the Figure to improve the readability.

A: We appreciate your suggestion! We added a coordinate system to the image as suggested.

Comment 9:

Due to the work presents a lot of statistical analysis from experimental data. It will be quite useful for the readership to extent the Statistical Analysis section. Instead of only set forth which software and methods are used, the authors should explain in detail why and how the statistical methods were selected.

A: We appreciate your comment about the lack of clarity in the statistical analysis section and agree with your comments.

In order to clarify the way in which statistical analysis was conducted, we rewrote the section on statistical methods giving more details of the computational tools used and statistical analysis.

We specified which STATS v.3.6.0 and LAWSTAT v3.2 packages of R were used.

We also clarified that two ANOVAS were conducted (one-way and two-way). One-way ANOVA was conducted to assess the influence on the defined control and experimental groups (PUERRF, SALINE23, SALINE5, DEIO23 and DEIO5) considering only one factor and one level. Subsequently, two-way ANOVA was used to assess 2 factors with two levels in each and two levels in each factor (deionized water or saline solution and room or refrigerated temperature). Thus, ANOVA only indicates that there is a difference in the groups under analysis, not indicating which group is different. Thus, the post-hoc Tukey HSD test was used to determine the divergent group.

We emphasize that the bootstrapping resampling method aims at ensuring that the hypotheses of application of ANOVA (normality and homoscedasticity) are met, and we have detailed how the bootstrapping method was defined and used by adding references to the text.

Comment 10:

In the caption of Figure 5, the variable is missing after the phrase “Performance index”. Moreover, the last sentence is difficult to read.

A: We appreciate the comment. The mathematical symbol used δ (delta) was corrupted by the submission platform. We checked the last section and there was an error in the translation. We have corrected the text.

Comment 11.

The sentence in line 194 has grammatical errors. Check spelling, grammar and typos along the whole manuscript.

A: We appreciate the comment and inform you that the entire manuscript has been sent back to a specialized translation service.

Comment 12.

In the Discussion, the ideas therein are certainly true; however, it shows some mixed concepts and unclear ideas. For instance,

1) Up to this point of the paper, the authors had not introduced neither linear nor non-lineal modeling of the electrical resistivity. It thus could confuse the readership owing that the work is supposed to be based on “linear” modeling by statistics.

It will be quite useful, again, to mention why they used linear statistical models instead of non-linear.

A: Using a non-linear approach is a great approach which was confirmed by the group in a previously published work. Unlike the non-linear approach, which allows to describe the impedance dynamics as a function of the procedure time and other dynamic parameters, here we seek to standardize notable points on the impedance curve. We believe that the present study is different because it allows an easy estimation without the need to use computational resources.

We agree with your suggestion to mention that it is a linear statistical model and highlight the contribution of this approach to the detriment of the non-linear approach and have included it at the end of the introduction.

References

1. Burton C. Heavy tailed distributions of effect sizes in systematic reviews of complex interventions. PLoS One. 2012;7. doi:10.1371/journal.pone.0034222

2. Bjerkli IH, Laurvik H, Nginamau ES, Søland TM, Costea D, Hov H, et al. Tumor budding score predicts lymph node status in oral tongue squamous cell carcinoma and should be included in the pathology report. PLoS One. 2020;15: 1–13. doi:10.1371/journal.pone.0239783

3. Wallisch C, Dunkler D, Rauch G, de Bin R, Heinze G. Selection of variables for multivariable models: Opportunities and limitations in quantifying model stability by resampling. Stat Med. 2020; 1–13. doi:10.1002/sim.8779

4. Zaryab M, Singh-Moon RP, Hendon CP. Robust classification of contact orientation between tissue and an integrated spectroscopy and radiofrequency ablation catheter. Diagnostic Ther Appl Light Cardiol. 2017;10042: 100420O. doi:10.1117/12.2251654

5. Isobe Y, Watanabe H, Yamazaki N, Lu X, Kobayashi Y, Miyashita T, et al. Real-time temperature control system based on the finite element method for liver radiofrequency ablation: Effect of the time interval on control. Conf Proc IEEE Eng Med Biol Soc. 2013;2013: 392–6. doi:10.1109/EMBC.2013.6609519

---

## [Editor Report · Decision Letter 2]

23 Dec 2020

Parametric evaluation of impedance curve in radiofrequency ablation: A quantitative description of the asymmetry and dynamic variation of impedance in bovine ex vivo model

PONE-D-20-20867R2

Dear Dr. Ronei Delfino da Fonseca,

We’re pleased to inform you that your manuscript has been judged scientifically suitable for publication and will be formally accepted for publication once it meets all outstanding technical requirements.

Kind regards,

Eugene Demidenko, Ph.D.

Academic Editor

PLOS ONE

Additional Editor Comments (optional):

The authors addressed the comments and the paper is in publishable form now.
---

## [Editor Report · Acceptance letter]

6 Jan 2021

PONE-D-20-20867R2 

Parametric evaluation of impedance curve in radiofrequency ablation: A quantitative description of the asymmetry and dynamic variation of impedance in bovine *ex vivo* model 

Dear Dr. da Fonseca:

I'm pleased to inform you that your manuscript has been deemed suitable for publication in PLOS ONE. Congratulations! Your manuscript is now with our production department. 

Kind regards, 

on behalf of

Dr. Eugene Demidenko 

Academic Editor

PLOS ONE